# Assessing the diagnostic accuracy of biochemical, anthropometric, and combined indices for metabolic syndrome prediction in a cohort from Qatar Biobank

**Muhammad Ammar Zahid**[1◉], **Abrar Abdelrahman**[1◉], **Hicham Raïq**[2], **Abdelhamid Kerkadi**[3], **Abdelali Agouni**[1]\*

**1** Department of Pharmaceutical Sciences, College of Pharmacy, QU Health, Qatar University, Doha, Qatar, **2** Department of Social Sciences, College of Arts and Sciences, Qatar University, Doha, Qatar, **3** Department of Diabetes Care & Patient Education, College of Health Sciences, University of Doha for Science and Technology (UDST), Doha, Qatar

◉ These authors contributed equally to this work
\* aagouni@qu.edu.qa

## Abstract

### Introduction

Metabolic syndrome (MetS) poses a substantial health risk, particularly in Qatar. This study aimed to compare the diagnostic accuracy of various indices for MetS identification in a well-characterized Qatari cohort from Qatar Biobank (QBB).

### Methods

This cross-sectional study included 692 adults (≥18 years) from the QBB, categorized into MetS and healthy groups using the International Diabetes Federation (IDF) criteria. We compared the distributions of biochemical, anthropometric, and combined indices between groups. Logistic regression assessed associations with MetS, adjusting for demographics. Receiver Operating Characteristic (ROC) analysis evaluated discriminative performance and identified optimal thresholds. Robustness was tested using a 75/25 train-test split. Stratified analyses examined the influence of age, gender, and nationality.

### Results

The MetS prevalence was 19.1% among participants. Individuals with MetS displayed significantly higher levels of all indices compared to the healthy group. Triglycerides (adjusted odd ratio (AOR): 4.93), waist circumference (AOR: 3.87), and lipid accumulation product (LAP) (AOR: 14.91) showed the strongest associations within their respective categories. LAP achieved the highest discriminative performance (area under the curve (AUC): 0.896; 95% CI: 0.870–0.923), followed by the

**Data availability statement:** Data are available from QBB under a controlled-access process and cannot be publicly shared due to IRB and data-transfer restrictions (IRB: Ex-2020-RES-ACC-0215-0125; Ex-2021-QF-QBB-RES-ACC-00049-0173). Requests should be directed to qphi@qf.org.qa.

**Funding:** The study was funded by the Qatar National Research Fund (Qatar Research Development and Innovation Council) [grant No. NPRP14S-0406-210150] and Qatar University [grants No. QUST-1-CPH-2025-247 and QUST-2-CPH-2021-221]. M.A.Z. is supported by a Ph.D. graduate assistantship from the Office of Graduate Studies (Qatar University). The statements made herein are solely the responsibility of the authors. The funders had no role in study design, data collection and analysis, decision to publish, or preparation of the manuscript.

**Competing interests:** The authors declare that they have no competing interests.

visceral adiposity index (VAI) (AUC: 0.877) and TyG × waist circumference (AUC: 0.872). LAP's optimal threshold was 37.1, with a sensitivity of 0.856 and a specificity of 0.789. Combined indices consistently outperformed individual measures. Discriminative accuracy was comparable across genders and nationalities but higher in individuals under 45 years.

## Conclusion

Combined indices, particularly LAP, demonstrate superior discriminative ability for MetS in this Qatari cohort. Incorporating LAP into routine clinical practice could improve MetS detection and facilitate timely interventions. Further validation in larger, diverse populations is, however, warranted.

## Introduction

Metabolic syndrome (MetS) is a multifactorial disorder defined by a cluster of metabolic abnormalities that increase the risk of atherosclerotic cardiovascular disease (ASCVD) and type 2 diabetes mellitus (T2DM) [1]. The 2009 Harmonized Criteria diagnose MetS based on central obesity—measured by ethnicity-specific waist circumference (WC)—plus at least two of the following: elevated triglycerides (TG), reduced HDL cholesterol, elevated blood pressure, or elevated fasting glucose [2]. Globally, MetS affects 12.5–31.4% of adults, with marked regional and socioeconomic variations [3,4].

In Qatar, recent evidence indicates a particularly high burden. A 2024 meta-analysis estimated an overall prevalence of 26%, higher among adults ≥40 years and when using International Diabetes Federation (IDF) criteria [5]. National surveys and primary care data report similar or higher rates (28–48.8%), especially among men and middle-aged adults [6,7]. These trends mirror high rates of key MetS components in the Qatari population, including hypertension (15.7%), dyslipidemia (11.1%), and obesity (35–37%) [8]. MetS contributes substantially to morbidity, mortality, and healthcare costs, accounting for millions of cardiovascular deaths globally and tripling annual costs among affected hypertensive patients [9,10]. Early and accurate identification of at-risk individuals is therefore essential for effective prevention.

Although multiple criteria exist, reliable identification of MetS remains challenging due to its heterogeneous etiology. Biochemical markers capture metabolic dysfunction but vary dynamically and lack clear diagnostic thresholds [11]. Anthropometric indices such as body mass index (BMI), WC, and waist-to-hip ratio (WHR) provide simple estimates of adiposity [12–15] but cannot distinguish fat distribution or composition. To address these limitations, composite indices—such as the Homeostatic Model Assessment of Insulin Resistance (HOMA-IR), Visceral Adiposity Index (VAI), Triglyceride-Glucose (TyG) index, and Lipid Accumulation Product (LAP)—have shown improved discriminative performance [16–18]. More recently Cholesterol, High-density lipoprotein, and Glucose (CHG) index have been used for the diagnosis

of diabetes mellitus and is associated with MetS [19,20]. However, the diagnostic accuracy and optimal cut-offs of these combined parameters remain unexamined in the Qatari population.

This study evaluates and compares the discriminative performance of biochemical, anthropometric, and combined indices for MetS within the Qatar Biobank (QBB) cohort to establish population-specific evidence and identify optimal diagnostic thresholds for early detection.

## Methods

### Study population

This cross-sectional study included Qatari nationals and long-term residents of Qatar (≥15 years) recruited from QBB, the first national population-based prospective cohort study established by Qatar Foundation, the Ministry of Public Health, and Imperial College London. QBB stores biological samples and data to study genetic, lifestyle, and environmental factors influencing disease incidence in Qatar [21]. Eligible participants were otherwise healthy male and female adults (≥ 18 years) with comprehensive metabolic status data available as detailed below. Individuals with life-threatening conditions such as cardiovascular diseases, cancer, autoimmune diseases, and chronic kidney or liver failure were excluded. Also excluded were pregnant and breastfeeding women. The metabolic syndrome status of the participant was determined using the IDF criteria [22]. Ethical approval was obtained from the institutional review board (IRB) of QBB (#Ex-2020-RES-ACC-0215–0125 and #Ex-2021-QF-QBB-RES-ACC-00049–0173) and Qatar University (#QU-IRB 1624-E/21), and all participants gave prior written informed consent to participate in the QBB program. The data were accessed for research purposes on 24 August 2022. The samples and data were deidentified by QBB before being provided to us, ensuring that authors did not have access to information that could identify individual participants during or after data collection.

### Biochemical, anthropometric, and combined measurements

Qualified healthcare personnel in the QBB clinic collected all clinical measurements and biological samples. Blood samples were obtained from each participant after overnight fasting. Standardized automated laboratory guidelines were followed to analyze certain clinical biomarkers including glucose (mmol/L), insulin (μIU/mL), hemoglobin A1c (%) (HbA1c), total cholesterol (mmol/L), HDL cholesterol (mmol/L), and TG (mmol/L). LDL cholesterol (mmol/L) levels were calculated using the Friedewald formula. Blood sample analyses were conducted in Hamad Medical Centre (HMC) Laboratory, Doha. Anthropometric measurements were performed using calibrated tools. Height (cm) and body weight (kg) were measured using a wall-mounted stadiometer and a scale (Seca, Hamburg, Germany), with participants being instructed to dress lightly and remove their footwear. WC (cm) was measured at the umbilical level above the iliac crest using a non-stretchable tape (Seca, Hamburg, Germany), and the WHR was calculated. Blood pressure measurements were taken with a mercury sphygmomanometer [20,23–27]. Combined measurements were calculated using the following equations.

$$VAI_{Male} = \frac{\text{Waist Circumference (cm)}}{39.68 + 1.88 \times \text{BMI (kg/m}^2)} \times \frac{\text{Triglycerides (mmol/L)}}{1.03} \times \frac{1.31}{\text{HDL Cholesterol (mmol/L)}}$$

$$VAI_{Female} = \frac{\text{Waist Circumference (cm)}}{36.88 + 1.89 \times \text{BMI (kg/m}^2)} \times \frac{\text{Triglycerides (mmol/L)}}{0.81} \times \frac{1.52}{\text{HDL Cholesterol (mmol/L)}}$$

$$LAP_{Male} = (\text{Waist Circumference (cm)} - 65) \times \text{Triglycerides (mmol/L)}$$

$$LAP_{Female} = (\text{Waist Circumference (cm)} - 58) \times \text{Triglycerides (mmol/L)}$$

$$AIP = \log_{10}\left(\frac{\text{Triglycerides (mmol/L)}}{\text{HDL Cholesterol (mmol/L)}}\right)$$

$$TyG = \ln\left(\frac{\text{Triglycerides (mg/dL)} \times \text{Glucose(mg/dL)}}{2}\right)$$

$$HOMA-IR = \frac{\text{Glucose (mmol/L)} \times \text{Insulin}(\mu U/mL)}{22.5}$$

$$CHG\ index = \ln\left(\frac{\text{Total Cholesterol (mg/dL)} \times \text{Fasting Glucose(mg/dL)}}{2 \times \text{HDL (mg/dL)}}\right)$$

## Statistical analysis

All data were processed and analyzed using R studio software (V 2024.09.0 + 375) running R (V 4.4.0). Descriptive statistics were reported as median and interquartile range (IQR) for continuous variables and frequencies and percentages for categorical variables. Normality was assessed using the Shapiro-Wilk test and visual inspection of histograms (S1 Fig). Between-group comparisons were performed using independent t-test and one-way ANOVA for normally distributed data and Kruskal–Wallis for non-normally distributed data. The Pearson correlation coefficient was used to assess the relationships between MetS and the different measurements. Logistic regression models were applied to analyze the association between MetS (independent variable) and biochemical, anthropometric, and combined measurements (dependent variables) using unadjusted models and models adjusted for age, nationality, and gender. For the logistic regression analysis, all continuous variables were standardized (converted to z-scores) to allow for the comparison of odds ratios across different measures. Predictive analysis was performed using Receiver Operating Characteristic (ROC) curve analysis, and the area under the curve (AUC) was calculated to assess the discriminatory power of each index. Optimal thresholds for each index were determined using Youden's index, which maximizes the sum of sensitivity and specificity. The robustness of the discriminatory models was evaluated using a 75/25 train-test split. DeLong's test was used to compare ROC curves.

## Results

### Demographics of the study population

The study included 692 participants, of whom 560 were healthy individuals and 132 were diagnosed with MetS. The participants were predominantly female (57%) and of Qatari (75%) origin. There was no statistically significant difference in the gender distribution (p = 0.2) and nationality (p = 0.055) between the two groups. The median age was significantly higher (P < 0.001) in the MetS group (47 years [IQR: 39–55]) compared to the healthy group (33 years [IQR: 27–42]). Biochemical and anthropometric measurements demonstrated statistically significant variations between the groups (p < 0.001). Participants with MetS exhibited markedly higher levels of glucose, insulin, HbA1c, total cholesterol, LDL cholesterol, and TG, accompanied by lower HDL cholesterol compared to healthy participants. Moreover, these participants had higher BMI, WC, and WHR medians. Combined Measures mainly the CHG Index, HOMA-IR, TyG Index, TG/HDL, TyG × BMI, TyG × WC, TyG × WHR, VAI, LAP, and AIP were also increased in the MetS as compared to the healthy individuals. The most predominant MetS condition identified in this cohort was central obesity, observed in 59% of the total population and 50% of the otherwise healthy population. Other conditions including high triglyceride, low HDL, high glucose, and high BP were similarly more prevalent in the MetS group than in the healthy group. In terms of the distribution of

the MetS conditions, the majority of individuals in the MetS group (73%) had at least two MetS conditions, whereas 20% and 7.6% had three and four conditions, respectively. On the other hand, around 96% of the healthy participants had no more than one MetS condition. These distributions validate the appropriate classification of participants into the respective study groups. The demographics of the study population are presented in **Table 1** and the differences of the biochemical, anthropometric, and combined indices in the two groups are shown in **Fig 1**.

The characteristics of the participants stratified by the number of MetS conditions (ranging from 0 to 4) showed significant differences in the indices under consideration (**Table 2**, **Fig 2**). Gender and nationality showed no significant differences across the strata (p = 0.8). The age distribution significantly varied with the number of MetS conditions, showing a progressive increase in the median age from 31 years in those with no MetS conditions to 48 years in those with four conditions (p < 0.001). All biochemical and anthropometric parameters demonstrated significant trends across the groups (p < 0.001). Notably, from zero to four MetS conditions, glucose increased from 4.70 to 6.00 mmol/L, insulin from 8 to 17 μIU/mL, and TG from 0.80 to 2.30 mmol/L. HDL levels showed an inverse trend, decreasing from 1.54 to 0.92 mmol/L. Similarly, BMI and WC increased from 26.5 to 31.3 kg/m² and 80 to 104 cm, respectively. The combined indices revealed statistically significant progressive increases across the groups (p < 0.001). Markers of insulin resistance, namely HOMA-IR and the TyG index, rose substantially, with HOMA-IR increasing from 1.59 to 4.45 and the TyG index from 8.03 to 9.22 between individuals with no conditions and those with four conditions. The CHG Index increased from 4.82 to 5.73 across the strata. VAI, LAP, and AIP indices showed similar trends, with VAI increasing from 0.74 to 3.53, LAP from 15 to 97, and AIP transitioning from −0.28 to 0.36 as the number of MetS conditions increased. These findings illustrate a pattern of worsening metabolic parameters corresponding to increasing MetS conditions, suggesting a cumulative effect of multiple metabolic abnormalities.

## Association between biochemical, anthropometric, and combined measures with MetS

Multivariate logistic regression analysis was performed to quantify the association between biochemical, anthropometric, and combined measures with MetS. The odds ratios are given for both adjusted and unadjusted models, with the adjusted model controlling for gender, nationality, and age group (**Table 3**). Among the biochemical measurements, TG showed the strongest association with MetS risk in both the unadjusted (OR: 4.97, 95% CI: 3.88–6.46) and adjusted (OR: 4.93, 95% CI: 3.79–6.52) models. Moreover, the other biochemical measurements, including glucose, insulin, and HbA1c, illustrated a positive association with the MetS risk with a slightly reduced OR in the adjusted model. Similarly, the anthropometric measurements also demonstrated a positive association with the MetS risk (**Fig 3**). Within this category, WC exhibited the strongest association with an odds ratio of 3.63 (95% CI: 2.99–4.46) in the unadjusted model, increasing to 3.87 (95% CI: 3.10–4.91) in the adjusted model. Building upon these individual marker findings, the analysis of combined measures revealed an even more pronounced association. Among these composite markers, LAP reflected the strongest association, with consistent ORs of 14.84 (95% CI: 10.51–21.50) in the unadjusted and 14.91 (95% CI: 10.25–22.33) in the adjusted models. The CHG index also showed a significant positive association with MetS (adjusted OR: 3.40, 95% CI: 2.80–4.18), although its predictive strength was more moderate compared to top-performing indices like LAP. Notably, LAP significantly outperformed other anthropometric and biochemical measurements, reflecting the highest association with MetS risk.

## Discriminative performance of the indices to predict MetS

The ROC curves and the AUC analysis illustrate the discriminative abilities of various indices to predict MetS (**Table 4**). Most of the indices demonstrated high accuracy in predicting MetS with AUC values ranging between 0.896 and 0.733. LAP showed the highest discriminative ability with an AUC of 0.896 (95% CI: 0.87–0.923). VAI and TyG×WC performed similarly with AUC values of 0.877 (95% CI: 0.843–0.911) and 0.872 (95% CI: 0.843–0.9), respectively. The TyG index, TG/HDL, and AIP showed good discriminative performance with AUC values above 0.85. The CHG Index also

**Table 1. Demographic and clinical characteristics of the study population by MetS status.**

| Characteristics | Overall (N = 692) | Healthy (N = 560) | MetS (N = 132) | p-value |
|---|---|---|---|---|
| **Gender** | | | | 0.2 |
| Female | 397 (57%) | 315 (56%) | 82 (62%) | |
| Male | 295 (43%) | 245 (44%) | 50 (38%) | |
| **Nationality** | | | | 0.055 |
| Non-Qatari | 175 (25%) | 133 (24%) | 42 (32%) | |
| Qatari | 517 (75%) | 427 (76%) | 90 (68%) | |
| **Age** | 36 (28, 45) | 33 (27, 42) | 47 (39, 55) | **<0.001** |
| **Biochemical Measurements** | | | | |
| Glucose (mmol/L) | 4.90 (4.60, 5.30) | 4.80 (4.50, 5.10) | 5.40 (4.98, 6.00) | **<0.001** |
| Insulin (µIU/mL) | 10 (7, 13) | 9 (6, 12) | 13 (10, 20) | **<0.001** |
| HbA1c (%) | 5.30 (5.00, 5.50) | 5.20 (5.00, 5.40) | 5.70 (5.40, 5.90) | **<0.001** |
| Total Chol. (mmol/L) | 4.80 (4.30, 5.40) | 4.80 (4.20, 5.30) | 5.10 (4.55, 5.80) | **<0.001** |
| HDL (mmol/L) | 1.35 (1.11, 1.63) | 1.43 (1.20, 1.70) | 1.10 (0.98, 1.24) | **<0.001** |
| LDL (mmol/L) | 2.90 (2.33, 3.44) | 2.83 (2.30, 3.38) | 3.20 (2.62, 3.81) | **<0.001** |
| TG (mmol/L) | 1.00 (0.70, 1.40) | 0.90 (0.70, 1.20) | 1.70 (1.20, 2.10) | **<0.001** |
| **Anthropometric Measurements** | | | | |
| BMI (kg/m²) | 28.6 (25.0, 32.6) | 27.7 (24.4, 31.7) | 31.1 (29.3, 36.3) | **<0.001** |
| WC (cm) | 87 (78, 96) | 83 (75, 92) | 97 (91, 105) | **<0.001** |
| WHR | 0.81 (0.74, 0.89) | 0.79 (0.73, 0.87) | 0.89 (0.83, 0.95) | **<0.001** |
| **Combined Measures** | | | | |
| CHG Index | 5.03 (4.78, 5.31) | 4.95 (4.73, 5.20) | 5.41 (5.24, 5.61) | **<0.001** |
| HOMA-IR | 2.12 (1.42, 3.04) | 1.87 (1.30, 2.64) | 3.20 (2.30, 4.87) | **<0.001** |
| TyG Index | 8.28 (7.96, 8.63) | 8.16 (7.87, 8.48) | 8.88 (8.60, 9.13) | **<0.001** |
| TG/HDL | 0.73 (0.48, 1.15) | 0.64 (0.43, 0.94) | 1.46 (1.03, 2.15) | **<0.001** |
| TyG × BMI | 241 (204, 275) | 228 (196, 263) | 280 (259, 318) | **<0.001** |
| TyG × WC | 714 (624, 820) | 678 (602, 776) | 866 (799, 954) | **<0.001** |
| TyG × WHR | 6.71 (5.98, 7.55) | 6.47 (5.85, 7.22) | 7.90 (7.11, 8.52) | **<0.001** |
| VAI | 1.10 (0.70, 1.76) | 0.95 (0.63, 1.41) | 2.33 (1.74, 3.32) | **<0.001** |
| LAP | 25 (14, 47) | 20 (12, 34) | 61 (45, 85) | **<0.001** |
| AIP | −0.14 (−0.32, 0.06) | −0.19 (−0.37, −0.03) | 0.16 (0.01, 0.33) | **<0.001** |
| **MetS conditions** | | | | |
| Central obesity | 411 (59%) | 279 (50%) | 132 (100%) | **<0.001** |
| High Triglyceride | 109 (16%) | 42 (7.5%) | 67 (51%) | **<0.001** |
| Low HDL | 192 (28%) | 103 (18%) | 89 (67%) | **<0.001** |
| High Glucose | 182 (26%) | 87 (16%) | 95 (72%) | **<0.001** |
| High B.P | 92 (13%) | 33 (5.9%) | 59 (45%) | **<0.001** |
| **Number of MetS conditions** | | | | |
| None | 324 (47%) | 324 (58%) | 0 (0%) | |
| One | 214 (31%) | 214 (38%) | 0 (0%) | |
| Two | 112 (16%) | 16 (2.9%) | 96 (73%) | |
| Three | 31 (4.5%) | 5 (0.9%) | 26 (20%) | |
| Four | 11 (1.6%) | 1 (0.2%) | 10 (7.6%) | |

Abbreviations: AIP, Atherogenic Index of Plasma; BMI, Body Mass Index; B.P, Blood Pressure; CHG Index, Cholesterol, High-density lipoprotein, and Glucose Index; Chol., Cholesterol; HbA1c, Hemoglobin A1c; HDL, High-Density Lipoprotein; HOMA-IR, Homeostatic Model Assessment of Insulin Resistance; LAP, Lipid Accumulation Product; LDL, Low-Density Lipoprotein; MetS, Metabolic Syndrome; TG, Triglycerides; TG/HDL, Triglyceride to High-Density Lipoprotein ratio; TyG Index, Triglyceride-Glucose Index; VAI, Visceral Adiposity Index; WC, Waist Circumference; WHR, Waist-to-Hip Ratio.

# Distribution of Indices by Metabolic Syndrome Status

Significance levels: **** p<0.0001, *** p<0.001, ** p<0.01, * p<0.05, ns p≥0.05

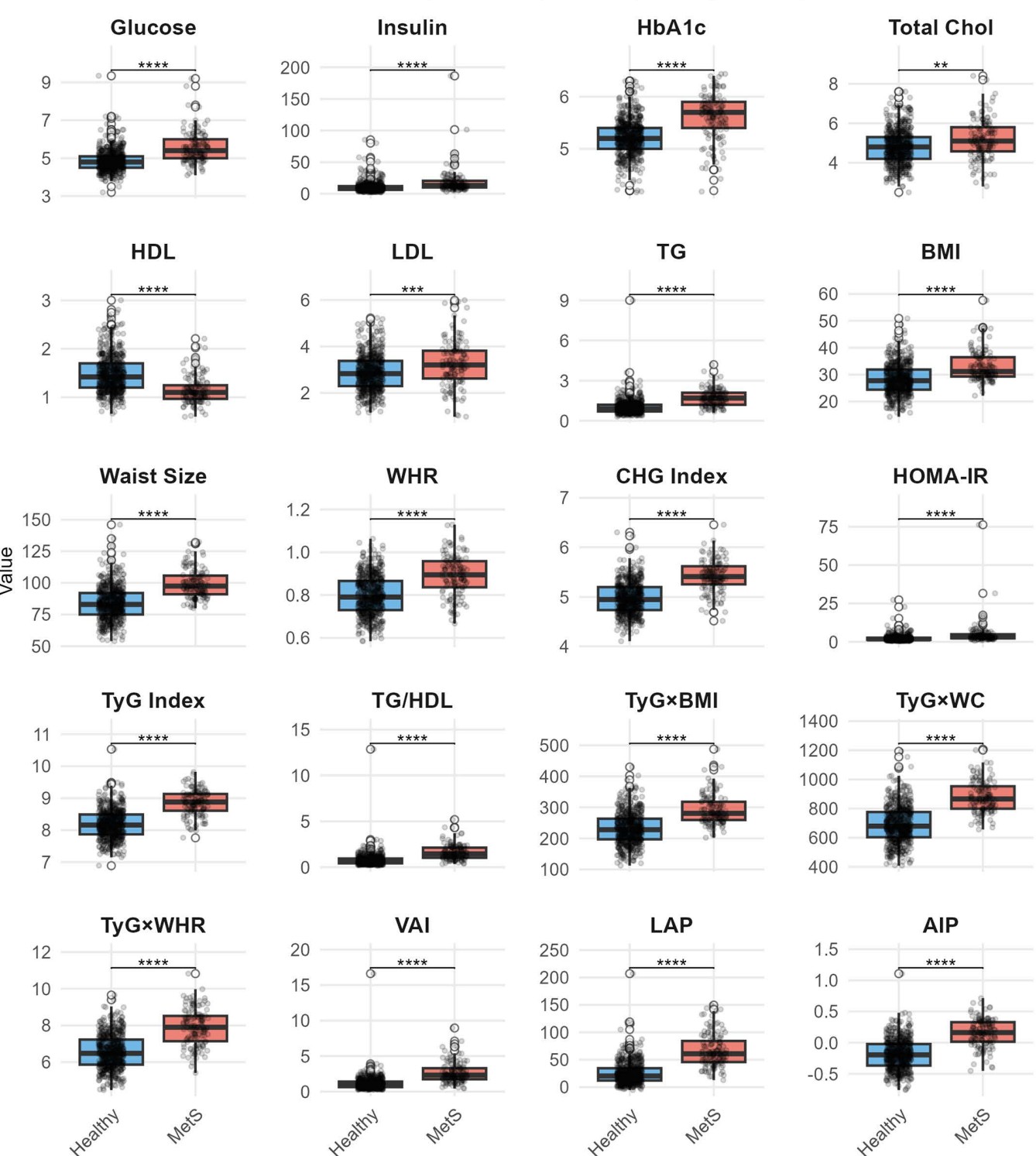

**Fig 1. Box plots of biochemical, anthropometric, and combined indices by MetS status.**

**Table 2. Demographic and clinical characteristics of the study population stratified by the number of MetS conditions.**

| Metabolic Syndrome | Overall, N = 692 | 0, N = 324 | 1, N = 214 | 2, N = 112 | 3, N = 31 | 4, N = 11 | p-value |
|---|---|---|---|---|---|---|---|
| **Gender** | | | | | | | 0.8 |
| Female | 397 (57%) | 182 (56%) | 128 (60%) | 66 (59%) | 15 (48%) | 6 (55%) | |
| Male | 295 (43%) | 142 (44%) | 86 (40%) | 46 (41%) | 16 (52%) | 5 (45%) | |
| **Nationality** | | | | | | | **0.050** |
| Non-Qatari | 175 (25%) | 71 (22%) | 52 (24%) | 36 (32%) | 13 (42%) | 3 (27%) | |
| Qatari | 517 (75%) | 253 (78%) | 162 (76%) | 76 (68%) | 18 (58%) | 8 (73%) | |
| Age | 36 (28, 45) | 31 (25, 38) | 36 (29, 47) | 45 (38, 52) | 50 (40, 57) | 48 (44, 57) | **<0.001** |
| **Biochemical Measurements** | | | | | | | |
| Glucose (mmol/L) | 4.90 (4.60, 5.30) | 4.70 (4.50, 5.00) | 5.00 (4.60, 5.42) | 5.20 (4.90, 5.70) | 5.80 (5.65, 6.30) | 6.00 (5.65, 6.35) | **<0.001** |
| Insulin (µIU/mL) | 10 (7, 13) | 8 (6, 10) | 11 (8, 14) | 12 (8, 17) | 15 (11, 23) | 17 (12, 24) | **<0.001** |
| HbA1c (%) | 5.30 (5.00, 5.50) | 5.20 (5.00, 5.40) | 5.30 (5.10, 5.60) | 5.55 (5.20, 5.80) | 5.80 (5.60, 6.05) | 5.80 (5.70, 6.10) | **<0.001** |
| Total Chol. (mmol/L) | 4.80 (4.30, 5.40) | 4.70 (4.20, 5.20) | 4.90 (4.30, 5.40) | 5.10 (4.59, 5.80) | 4.90 (4.40, 5.85) | 5.50 (4.80, 5.95) | **<0.001** |
| HDL (mmol/L) | 1.35 (1.11, 1.63) | 1.54 (1.34, 1.85) | 1.23 (1.10, 1.52) | 1.10 (0.95, 1.27) | 1.10 (1.00, 1.23) | 0.92 (0.88, 1.00) | **<0.001** |
| LDL (mmol/L) | 2.90 (2.33, 3.44) | 2.70 (2.21, 3.13) | 3.05 (2.40, 3.53) | 3.25 (2.79, 3.82) | 2.95 (2.48, 3.76) | 3.38 (2.77, 4.01) | **<0.001** |
| TG (mmol/L) | 1.00 (0.70, 1.40) | 0.80 (0.60, 1.00) | 1.10 (0.80, 1.40) | 1.50 (1.20, 2.00) | 2.00 (1.70, 2.40) | 2.30 (2.00, 2.55) | **<0.001** |
| **Anthropometric Measurements** | | | | | | | |
| BMI (kg/m²) | 28.6 (25.0, 32.6) | 26.5 (23.2, 30.2) | 29.8 (26.4, 33.9) | 30.3 (27.9, 33.8) | 32.0 (28.7, 36.5) | 31.3 (30.8, 35.6) | **<0.001** |
| WC (cm) | 87 (78, 96) | 80 (71, 87) | 90 (81, 98) | 95 (88, 102) | 96 (92, 107) | 104 (97, 112) | **<0.001** |
| WHR | 0.81 (0.74, 0.89) | 0.77 (0.71, 0.84) | 0.83 (0.76, 0.89) | 0.89 (0.82, 0.94) | 0.88 (0.84, 0.96) | 0.92 (0.83, 0.98) | **<0.001** |
| **Combined Measures** | | | | | | | |
| CHG Index | 5.03 (4.78, 5.31) | 4.82 (4.65, 5.00) | 5.16 (4.96, 5.35) | 5.37 (5.19, 5.56) | 5.52 (5.36, 5.64) | 5.73 (5.51, 5.92) | **<0.001** |
| HOMA-IR | 2.12 (1.42, 3.04) | 1.59 (1.17, 2.22) | 2.40 (1.67, 3.22) | 2.79 (2.12, 4.01) | 3.58 (2.79, 6.22) | 4.45 (3.21, 6.65) | **<0.001** |
| TyG Index | 8.28 (7.96, 8.63) | 8.03 (7.78, 8.27) | 8.40 (8.10, 8.66) | 8.80 (8.47, 9.04) | 9.13 (8.90, 9.34) | 9.22 (9.17, 9.47) | **<0.001** |
| TG/HDL | 0.73 (0.48, 1.15) | 0.52 (0.37, 0.71) | 0.84 (0.62, 1.18) | 1.30 (0.93, 2.04) | 1.70 (1.47, 2.13) | 2.30 (1.96, 2.90) | **<0.001** |
| TyG × BMI | 241 (204, 275) | 212 (184, 244) | 251 (223, 288) | 268 (248, 301) | 283 (266, 332) | 292 (284, 327) | **<0.001** |
| TyG × WC | 714 (624, 820) | 642 (567, 707) | 762 (658, 835) | 827 (775, 920) | 869 (839, 979) | 985 (918, 1,018) | **<0.001** |
| TyG × WHR | 6.71 (5.98, 7.55) | 6.16 (5.63, 6.72) | 6.91 (6.30, 7.58) | 7.78 (7.08, 8.42) | 8.00 (7.51, 8.82) | 8.74 (7.72, 9.19) | **<0.001** |
| VAI | 1.10 (0.70, 1.76) | 0.74 (0.52, 1.04) | 1.32 (0.96, 1.80) | 2.17 (1.47, 2.92) | 2.49 (2.05, 3.11) | 3.53 (3.29, 4.09) | **<0.001** |
| LAP | 25 (14, 47) | 15 (8, 24) | 30 (19, 50) | 51 (36, 67) | 62 (54, 92) | 97 (81, 120) | **<0.001** |
| AIP | −0.14 (−0.32, 0.06) | −0.28 (−0.43, −0.15) | −0.08 (−0.20, 0.07) | 0.11 (−0.03, 0.31) | 0.23 (0.16, 0.33) | 0.36 (0.29, 0.46) | **<0.001** |

Abbreviations: AIP, Atherogenic Index of Plasma; BMI, Body Mass Index; B.P, Blood Pressure; CHG Index, Cholesterol, High-density lipoprotein, and Glucose Index; Chol., Cholesterol; HbA1c, Hemoglobin A1c; HDL, High-Density Lipoprotein; HOMA-IR, Homeostatic Model Assessment of Insulin Resistance; LAP, Lipid Accumulation Product; LDL, Low-Density Lipoprotein; MetS, Metabolic Syndrome; TG, Triglycerides; TG/HDL, Triglyceride to High-Density Lipoprotein ratio; TyG Index, Triglyceride-Glucose Index; VAI, Visceral Adiposity Index; WC, Waist Circumference; WHR, Waist-to-Hip Ratio.

demonstrated strong performance with an AUC of 0.842 (95% CI: 0.805–0.878). Anthropometric measures (WC and BMI) demonstrated moderate discriminative ability (AUC = 0.816 and 0.739, respectively), while the biochemical parameters (glucose and HbA1c) showed fair discriminative performance (AUC = 0.766 and 0.76, respectively). BMI showed the highest sensitivity (0.932), whereas VAI showed the highest specificity (0.855). Notably, LAP provided the most balanced performance (sensitivity: 0.856, specificity: 0.789) at a threshold of 37.1 highlighting its potential value as a primary screening tool in clinical practice. While the optimal threshold for each index provides a valuable decision point for classifying individuals as having or not having metabolic syndrome, it is important to note that this threshold is context-dependent and must be selected based on the specific clinical application and the relative importance of sensitivity and specificity. For instance,

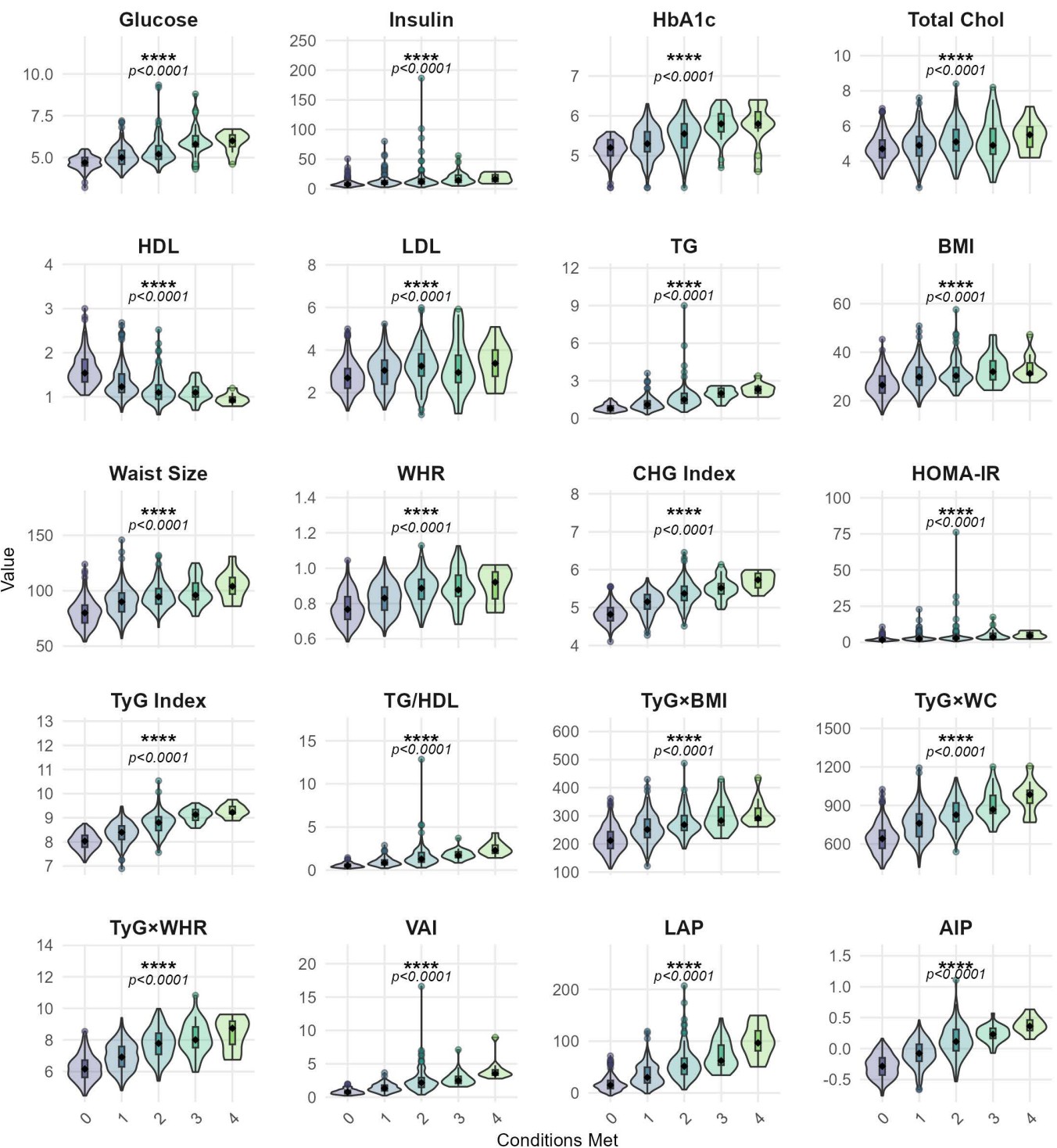

**Distribution of Metabolic Indices by Number of MetS Conditions**

Sample sizes: Condition 0 (n=324), Condition 1 (n=214), Condition 2 (n=112), Condition 3 (n=31), Condition 4 (n=11)

**Fig 2. Distribution of indices by the number of MetS conditions.**

**Table 3. Multivariate logistic regression analysis of MetS risk associated with biochemical, anthropometric, and combined indices.**

| Marker | Unadjusted OR (95% CI) | P-value | Adjusted OR (95% CI) | p-value |
|---|---|---|---|---|
| **Biochemical Measures** | | | | |
| Glucose (mmol/L) | 1.99 (1.71-2.33) | <0.001 | 1.74 (1.49-2.05) | <0.001 |
| Insulin (µIU/mL) | 2.41 (1.69-3.61) | <0.001 | 2.19 (1.54-3.28) | <0.001 |
| HbA1c (%) | 2.43 (2.08-2.86) | <0.001 | 2.09 (1.77-2.48) | <0.001 |
| Total Chol. (mmol/L) | 1.10 (0.96-1.25) | 0.172 | 1.10 (0.95-1.26) | 0.195 |
| HDL (mmol/L) | 0.31 (0.25-0.37) | <0.001 | 0.21 (0.16-0.27) | <0.001 |
| LDL (mmol/L) | 1.08 (0.95-1.23) | 0.262 | 1.12 (0.98-1.29) | 0.108 |
| TG (mmol/L) | 4.97 (3.88-6.46) | <0.001 | 4.93 (3.79-6.52) | <0.001 |
| **Anthropometric Measures** | | | | |
| BMI (kg/m²) | 2.31 (1.97-2.73) | <0.001 | 2.16 (1.82-2.58) | <0.001 |
| WC (cm) | 3.63 (2.99-4.46) | <0.001 | 3.87 (3.10-4.91) | <0.001 |
| WHR | 2.61 (2.22-3.09) | <0.001 | 3.16 (2.56-3.95) | <0.001 |
| **Combined Measures** | | | | |
| HOMA-IR | 2.95 (2.08-4.35) | <0.001 | 2.41 (1.71-3.54) | <0.001 |
| TyG Index | 4.27 (3.50-5.26) | <0.001 | 4.14 (3.33-5.21) | <0.001 |
| TG/HDL | 4.51 (3.58-5.76) | <0.001 | 5.31 (4.09-7.02) | <0.001 |
| VAI | 7.65 (5.79-10.32) | <0.001 | 7.14 (5.33-9.75) | <0.001 |
| LAP | 14.84 (10.51-21.50) | <0.001 | 14.91 (10.25-22.33) | <0.001 |
| AIP | 4.61 (3.76-5.71) | <0.001 | 5.76 (4.52-7.47) | <0.001 |
| TyG×BMI | 4.19 (3.40-5.23) | <0.001 | 3.76 (3.03-4.73) | <0.001 |
| TyG×WC | 6.15 (4.85-7.94) | <0.001 | 7.83 (5.86-10.69) | <0.001 |
| TyG×WHR | 3.97 (3.29-4.84) | <0.001 | 5.56 (4.33-7.26) | <0.001 |
| CHG Index | 3.33 (2.79-4.01) | <0.001 | 3.40 (2.80-4.18) | <0.001 |

Abbreviations: AIP, Atherogenic Index of Plasma; BMI, Body Mass Index; B.P, Blood Pressure; CHG Index, Cholesterol, High-density lipoprotein, and Glucose Index; Chol., Cholesterol; CI, Confidence Interval, HbA1c, Hemoglobin A1c; HDL, High-Density Lipoprotein; HOMA-IR, Homeostatic Model Assessment of Insulin Resistance; LAP, Lipid Accumulation Product; LDL, Low-Density Lipoprotein; MetS, Metabolic Syndrome; OR, Odds Ratio; TG, Triglycerides; TG/HDL, Triglyceride to High-Density Lipoprotein ratio; TyG Index, Triglyceride-Glucose Index; VAI, Visceral Adiposity Index; WC, Waist Circumference; WHR, Waist-to-Hip Ratio.

Adjusted models include: Gender, Nationality, and Age Group – All continuous variables were standardized before analysis

the optimal threshold for WC was 86.5 cm, offering high sensitivity (0.909) but lower specificity (0.596), while HbA1c had an optimal threshold of 5.55%, yielding lower sensitivity (0.621) but higher specificity (0.843). These findings highlight the importance of considering the optimal threshold in conjunction with other performance metrics when choosing an appropriate index for predicting metabolic syndrome.

Pairwise comparisons of AUCs using DeLong's test revealed several statistically significant differences (p < 0.05) between the evaluated indices for predicting MetS (S1 Table). The LAP, with an AUC of 0.896, demonstrated significantly higher discriminative performance compared to TyG×WC (p = 0.0010), TyG×BMI (p < 0.0001), WC (p < 0.0001), BMI (p < 0.0001), WHR (p < 0.0001), and all basic measures considered together (p < 0.0001). Similarly, the VAI (AUC = 0.877) performed significantly better than TyG×BMI (p = 0.0106), WC (p = 0.0052), all traditional measures (p < 0.001), AIP (p = 0.0028), and TG/HDL (p = 0.0033). TyG×WC (AUC = 0.872) also showed statistically significant superiority over TyG×BMI (p = 0.0001), TyG×WHR (p = 0.0081), and all traditional measures (p < 0.001).

Among the traditional measurements, WC demonstrated significantly better discrimination than both BMI (p < 0.0001) and WHR (p = 0.0016). Furthermore, TG differed significantly from both glucose (p = 0.0146) and HDL (p = 0.0109), and insulin differed significantly from HOMA-IR (p < 0.0001). Finally, several other combined indices, including the TyG Index,

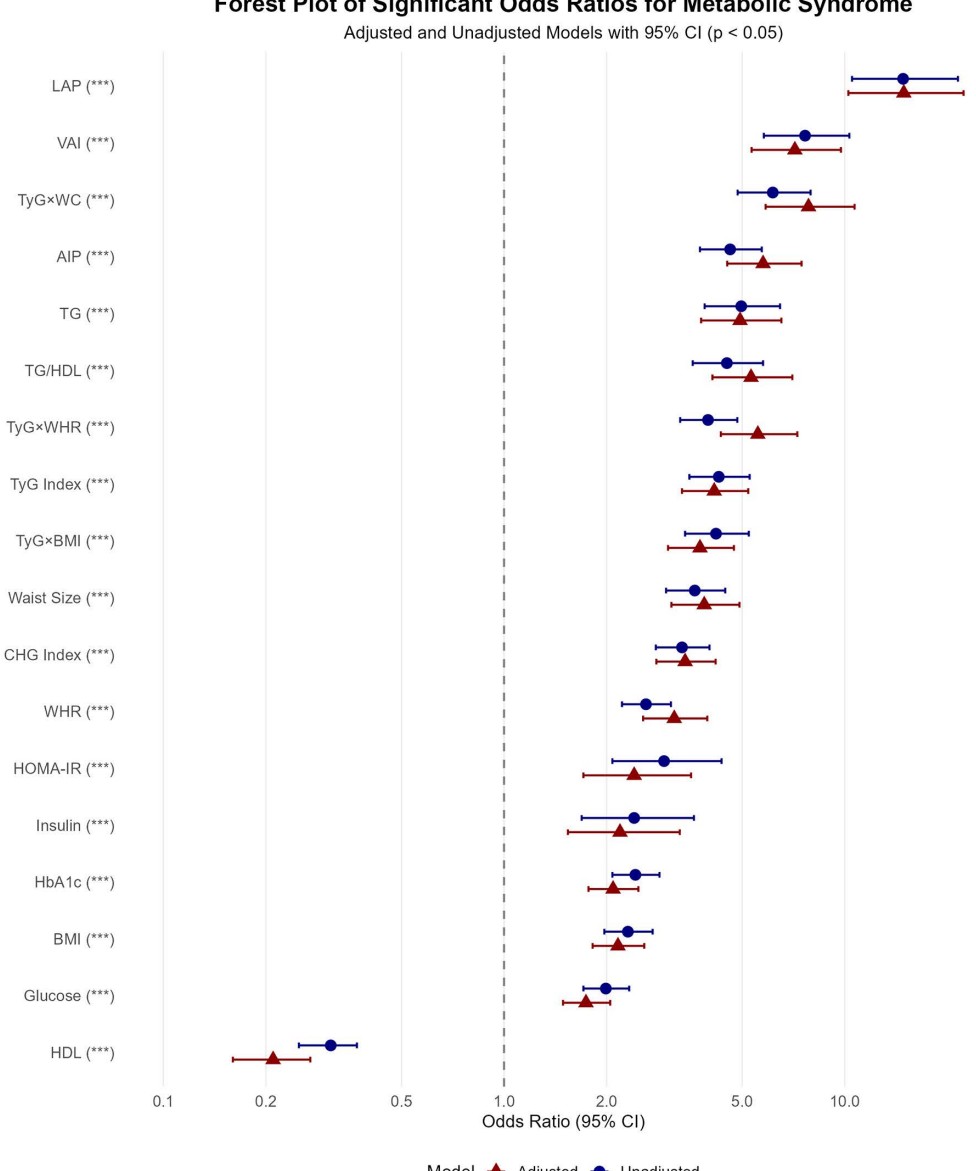

**Fig 3. Forest plot of odds ratios for the association of various indices with MetS, unadjusted (blue) and adjusted (red) for age, gender, and nationality.**

TG/HDL, and AIP, all showed significantly better performance compared to the traditional measures (p < 0.001 for all comparisons). Specifically, TG/HDL also outperformed BMI (p < 0.0001), WHR (p < 0.0001), and glucose (p = 0.0016) individually. These results suggest that combined indices, particularly LAP and VAI, offer superior discriminative performance for metabolic syndrome compared to individual biometric or biochemical measures.

## Sub-group analysis and the comparison of discriminative performance

A stratified analysis was conducted to examine the discriminative performance of various indices across different demographic subgroups (**Table 5**). In the gender-based analysis, while the point estimates of AUC values varied slightly

**Table 4. Discriminative performance (AUC, sensitivity, specificity) and optimal thresholds of various indices for MetS.**

| Index | AUC | 95% CI | Sensitivity | Specificity | Optimal threshold |
|---|---|---|---|---|---|
| **Biochemical Measures** | | | | | |
| TG (mmol/L) | 0.836 | 0.8-0.873 | 0.682 | 0.827 | 1.390 |
| HDL (mmol/L) | 0.777 | 0.734-0.821 | 0.773 | 0.671 | 1.275 |
| Glucose (mmol/L) | 0.766 | 0.722-0.811 | 0.697 | 0.700 | 5.050 |
| HbA1c (%) | 0.760 | 0.708-0.813 | 0.621 | 0.843 | 5.550 |
| Insulin (µIU/mL) | 0.733 | 0.687-0.779 | 0.771 | 0.590 | 9.850 |
| **Anthropometric Measures** | | | | | |
| WC | 0.816 | 0.781-0.85 | 0.909 | 0.596 | 86.500 |
| WHR | 0.753 | 0.707-0.799 | 0.742 | 0.655 | 0.838 |
| BMI | 0.739 | 0.698-0.779 | 0.932 | 0.477 | 27.375 |
| **Combined Measures** | | | | | |
| LAP | 0.896 | 0.87-0.923 | 0.856 | 0.789 | 37.100 |
| VAI | 0.877 | 0.843-0.911 | 0.758 | 0.855 | 1.738 |
| TyG×WC | 0.872 | 0.843-0.9 | 0.886 | 0.713 | 763.720 |
| TyG Index | 0.866 | 0.833-0.899 | 0.742 | 0.848 | 8.625 |
| TG/HDL | 0.857 | 0.823-0.892 | 0.765 | 0.805 | 1.015 |
| AIP | 0.857 | 0.822-0.892 | 0.765 | 0.807 | 0.007 |
| CHG Index | 0.842 | 0.805-0.878 | 0.742 | 0.827 | 5.275 |
| TyG×WHR | 0.833 | 0.796-0.87 | 0.667 | 0.830 | 7.505 |
| TyG×BMI | 0.818 | 0.785-0.851 | 0.902 | 0.634 | 245.062 |
| HOMA-IR | 0.769 | 0.726-0.812 | 0.679 | 0.753 | 2.655 |

Abbreviations: AIP, Atherogenic Index of Plasma; AUC, Area Under the Curve; BMI, Body Mass Index; CHG Index, Cholesterol, High-density lipoprotein, and Glucose Index; CI, Confidence Interval; HbA1c, Hemoglobin A1c; HDL, High-Density Lipoprotein; HOMA-IR, Homeostatic Model Assessment of Insulin Resistance; LAP, Lipid Accumulation Product; TG, Triglycerides; TG/HDL, Triglyceride to High-Density Lipoprotein ratio; TyG, Triglyceride-Glucose; VAI, Visceral Adiposity Index; WC, Waist Circumference; WHR, Waist-to-Hip Ratio.

AUC values range from 0 to 1, with 1 indicating perfect discrimination. Optimal thresholds were determined using Youden's J statistic.

between females and males, no statistically significant differences were found. Notably, LAP, VAI, TyG×WC, AIP, TG/HDL, and the CHG index maintained high AUC values in both genders, with slight numerical differences.

Similar trends were observed in the nationality-based analysis, where no statistically significant differences were detected between Qatari and non-Qatari participants, although the combined indices showed a numerically better performance in the non-Qatari group. However, a more pronounced difference in discriminative performance was observed across different age groups, with all indices demonstrating significantly better performance in individuals younger than 45 years compared to those 45 years and older (**Table 6**). Specifically, indices like VAI, LAP, AIP, TG/HDL and CHG index showed substantially higher AUC values in the younger age group (AUCs > 0.9) compared to the older group (AUCs around 0.8). The analysis of the ROC curves also confirmed these trends, showing statistically significant differences between the two age groups for all tested metrics. Specifically, differences were noted in the AUCs for WC, HDL, CHG index, VAI, TyG-WC, LAP, TG/HDL, AIP, TyG-WHR, WHR, TyG-BMI, TG, TyG, and BMI (all p < 0.01 except the last 4 p < 0.05). These results suggest a reduction in the discriminative capacity of these markers for MetS with advancing age, with a statistically significant difference found in all studied markers, indicating that the indices generally performed better in the younger age group. In summary, while discriminative performance appears robust across gender and nationality, age is a significant factor influencing the effectiveness of these indices, particularly the combined indices.

**Table 5. Stratified analysis of AUC values for various indices by gender, nationality, and age group.**

**Gender-based analysis**

| Index | Female (N = 397) | 95% CI | Male (N = 295) | 95% CI |
|---|---|---|---|---|
| LAP | 0.906 | 0.875-0.937 | 0.898 | 0.853-0.943 |
| VAI | 0.899 | 0.861-0.937 | 0.847 | 0.785-0.908 |
| TyG × WC | 0.892 | 0.860-0.924 | 0.895 | 0.852-0.939 |
| AIP | 0.890 | 0.851-0.929 | 0.845 | 0.784-0.906 |
| TG/HDL | 0.890 | 0.851-0.929 | 0.846 | 0.785-0.906 |
| TyG Index | 0.869 | 0.825-0.913 | 0.883 | 0.841-0.925 |
| CHG Index | 0.867 | 0.823-0.911 | 0.828 | 0.767-0.889 |
| TyG × WHR | 0.864 | 0.821-0.907 | 0.858 | 0.797-0.918 |
| WC | 0.842 | 0.803-0.881 | 0.831 | 0.776-0.886 |
| HDL | 0.840 | 0.794-0.885 | 0.713 | 0.627-0.800 |
| TG | 0.836 | 0.788-0.885 | 0.859 | 0.809-0.908 |

**Nationality-Based Analysis**

| Index | Qatari (N = 517) | 95% CI | Non-Qatari (N = 175) | 95% CI |
|---|---|---|---|---|
| LAP | 0.890 | 0.859-0.922 | 0.912 | 0.865-0.959 |
| VAI | 0.872 | 0.830-0.914 | 0.886 | 0.827-0.944 |
| TyG × WC | 0.868 | 0.835-0.901 | 0.886 | 0.832-0.939 |
| TyG Index | 0.857 | 0.816-0.899 | 0.877 | 0.823-0.931 |
| AIP | 0.852 | 0.808-0.895 | 0.863 | 0.804-0.923 |
| TG/HDL | 0.852 | 0.808-0.895 | 0.863 | 0.804-0.923 |
| CHG Index | 0.836 | 0.79-0.882 | 0.854 | 0.794-0.915 |
| TG | 0.829 | 0.783-0.875 | 0.846 | 0.784-0.907 |
| TyG × WHR | 0.829 | 0.785-0.873 | 0.845 | 0.776-0.915 |

**Age Group-Based Analysis (AUC Values)**

| Index | Age < 45 (N = 505) | 95% CI | Age ≥ 45 (N = 187) | 95% CI |
|---|---|---|---|---|
| VAI | 0.936 | 0.910-0.961 | 0.802 | 0.735-0.869 |
| LAP | 0.935 | 0.911-0.959 | 0.811 | 0.748-0.875 |
| AIP | 0.915 | 0.884-0.946 | 0.783 | 0.714-0.851 |
| TG/HDL | 0.915 | 0.884-0.946 | 0.783 | 0.714-0.851 |
| TyG × WC | 0.911 | 0.882-0.940 | 0.769 | 0.701-0.837 |
| CHG Index | 0.896 | 0.857-0.935 | 0.736 | 0.662-0.809 |
| TyG Index | 0.893 | 0.854-0.933 | 0.788 | 0.721-0.855 |
| HDL | 0.880 | 0.841-0.919 | 0.722 | 0.646-0.797 |
| TyG × WHR | 0.876 | 0.828-0.924 | 0.725 | 0.651-0.799 |

Abbreviations: AIP, Atherogenic Index of Plasma; AUC, Area Under the Curve; CHG Index, Cholesterol, High-density lipoprotein, and Glucose Index; CI, Confidence Interval; HDL, High-Density Lipoprotein; LAP, Lipid Accumulation Product; N, number of participants; TG, Triglycerides; TG/HDL, Triglyceride to High-Density Lipoprotein ratio; TyG, Triglyceride-Glucose; VAI, Visceral Adiposity Index; WC, Waist Circumference; WHR, Waist-to-Hip Ratio.

Only indices with AUC > 0.7 are shown for clarity. All results have p-value < 0.001 * Higher AUC indicates better discriminative ability * Indices generally performed better in: – Female vs Male subjects – Non-Qatari vs Qatari subjects – Age < 45 vs Age ≥ 45 groups.

## Training and test performance

The performance of various indices for predicting MetS was evaluated using a 75/25% train/test split to assess robustness. The results demonstrate the discriminative capability of these indices on both training and unseen testing data. Among the biochemical measures, TG level had the best performance on the test data (AUC = 0.864, Sensitivity: 0.606,

**Table 6. Pairwise comparisons of AUC values between age groups (<45 vs. ≥45) for various indices.**

| Metric | Age < 45 (AUC) | Age ≥ 45 (AUC) | Difference | Significance |
|---|---|---|---|---|
| WC | 0.861 | 0.697 | +0.164 | ** |
| HDL | 0.880 | 0.722 | +0.158 | ** |
| CHG Index | 0.896 | 0.736 | +0.160 | ** |
| VAI | 0.936 | 0.802 | +0.134 | ** |
| TyG-WC | 0.911 | 0.769 | +0.142 | ** |
| LAP | 0.935 | 0.811 | +0.123 | ** |
| TG/HDL | 0.915 | 0.783 | +0.132 | ** |
| AIP | 0.915 | 0.783 | +0.132 | ** |
| TyG-WHR | 0.876 | 0.725 | +0.151 | ** |
| WHR | 0.800 | 0.648 | +0.153 | ** |
| TyG-BMI | 0.846 | 0.720 | +0.125 | ** |
| TG | 0.872 | 0.759 | +0.114 | * |
| TyG | 0.893 | 0.788 | +0.105 | * |
| BMI | 0.762 | 0.643 | +0.119 | * |

Significance codes: ** p < 0.01, * p < 0.05

Abbreviations: AIP, Atherogenic Index of Plasma; AUC, Area Under the Curve; CHG Index, Cholesterol, High-density lipoprotein, and Glucose Index; CI, Confidence Interval; HDL, High-Density Lipoprotein; LAP, Lipid Accumulation Product; N, number of participants; TG, Triglycerides; TG/HDL, Triglyceride to High-Density Lipoprotein ratio; TyG, Triglyceride-Glucose; VAI, Visceral Adiposity Index; WC, Waist Circumference; WHR, Waist-to-Hip Ratio.

Specificity: 0.893, Accuracy: 0.838) followed by HDL (AUC = 0.834, Sensitivity:0.182, Specificity:0.336 Accuracy:0.306) and HbA1c (AUC = 0.774, Sensitivity:0.727, Specificity: 0.743, Accuracy: 0.740). Glucose, insulin, total cholesterol, and LDL cholesterol performed relatively poorly (AUC = 0.595 to 0.809) on the test data (**Table 7**, S4 Fig). The optimal cutoff values for each measure were selected based on ROC analysis of the whole dataset. When considering the anthropometric measures, the WHR performed best on the test data set with an AUC of 0.796 and a test accuracy of 0.671 followed by WC (AUC: 0.822, Test Accuracy: 0.630) while BMI had the worst performance (AUC: 0.735, Test Accuracy: 0.520) (**Table 7**, S5 Fig). The combined measures generally showed stronger discriminative capacity than the individual measures. The LAP demonstrated the best performance on the test set with an AUC of 0.905, a test sensitivity of 0.879, a test specificity of 0.779, and a test accuracy of 0.798, followed closely by the VAI with a test AUC of 0.889, a test sensitivity of 0.788, a test specificity of 0.779 and a test accuracy of 0.780. The CHG Index also performed very well on the test data, with a test AUC of 0.876 and a high test accuracy of 0.821. Several of the TyG-based combined indices (TyG Index, TG/HDL Ratio, TyG × WC, and TyG × WHR) also showed good performance, with test AUCs in the range of 0.864–0.890 and test accuracies in the range of 0.734–0.844. HOMA-IR showed reasonable performance, with a test AUC of 0.832 and a test accuracy of 0.776 (**Table 7**, **Fig 4**). Overall, these results indicate that the combined indices, particularly LAP and VAI, exhibited the most robust performance, with good discriminative power even in the unseen test data, indicating their potential as useful markers for predicting MetS. The individual biochemical measures generally had lower AUCs and accuracies compared to the combined indices in both train and test data. Further, the test sensitivity and specificity for many individual measures were not optimal.

## Discussion

This cross-sectional study examined the diagnostic accuracy of various biochemical, anthropometric, and combined indices for predicting MetS in the Qatari population. Our study revealed distinct differences in the metabolic profiles of healthy individuals and those with MetS. MetS patients showed significantly elevated biochemical markers, anthropometric measurements, and combined metabolic indices compared to healthy individuals. Notably, TG showed the strongest

**Table 7. Performance of indices in predicting MetS on training and test datasets.**

| Measure | Training AUC | Test AUC | Optimal cutoff | Test sensitivity | Test specificity | Test PPV | Test NPV | Test accuracy |
|---|---|---|---|---|---|---|---|---|
| **Biochemical Measures** | | | | | | | | |
| Glucose (mmol/L) | 0.780 | 0.724 | 5.05 | 0.636 | 0.700 | 0.333 | 0.891 | 0.688 |
| Insulin (µIU/mL) | 0.707 | 0.809 | 9.85 | 0.848 | 0.606 | 0.341 | 0.943 | 0.653 |
| HbA1c (%) | 0.756 | 0.774 | 5.45 | 0.727 | 0.743 | 0.400 | 0.920 | 0.740 |
| Total Chol. (mmol/L) | 0.613 | 0.595 | 4.96 | 0.485 | 0.607 | 0.225 | 0.833 | 0.584 |
| HDL (mmol/L) | 0.759 | 0.834 | 1.27 | 0.182 | 0.336 | 0.061 | 0.635 | 0.306 |
| LDL (mmol/L) | 0.613 | 0.642 | 3.00 | 0.606 | 0.614 | 0.270 | 0.869 | 0.613 |
| TG (mmol/L) | 0.828 | 0.864 | 1.45 | 0.606 | 0.893 | 0.571 | 0.906 | 0.838 |
| **Anthropometric Measures** | | | | | | | | |
| BMI (kg/m²) | 0.741 | 0.735 | 27.02 | 0.939 | 0.421 | 0.277 | 0.967 | 0.520 |
| WC (cm) | 0.813 | 0.822 | 86.50 | 0.879 | 0.571 | 0.326 | 0.952 | 0.630 |
| WHR | 0.740 | 0.796 | 0.84 | 0.818 | 0.636 | 0.346 | 0.937 | 0.671 |
| **Combined Measures** | | | | | | | | |
| CHG Index | 0.830 | 0.876 | 5.27 | 0.758 | 0.836 | 0.521 | 0.936 | 0.821 |
| HOMA-IR | 0.748 | 0.832 | 2.66 | 0.758 | 0.781 | 0.455 | 0.930 | 0.776 |
| TyG Index | 0.859 | 0.890 | 8.62 | 0.788 | 0.857 | 0.565 | 0.945 | 0.844 |
| TG/HDL Ratio | 0.846 | 0.888 | 1.02 | 0.788 | 0.807 | 0.491 | 0.942 | 0.803 |
| TyG × BMI | 0.823 | 0.805 | 245.06 | 0.848 | 0.621 | 0.346 | 0.946 | 0.665 |
| TyG × WC | 0.870 | 0.875 | 763.72 | 0.848 | 0.707 | 0.406 | 0.952 | 0.734 |
| TyG × WHR | 0.823 | 0.864 | 7.09 | 0.818 | 0.736 | 0.422 | 0.945 | 0.751 |
| VAI | 0.872 | 0.889 | 1.44 | 0.788 | 0.779 | 0.456 | 0.940 | 0.780 |
| LAP | 0.893 | 0.905 | 37.10 | 0.879 | 0.779 | 0.483 | 0.965 | 0.798 |
| AIP | 0.846 | 0.888 | 0.01 | 0.788 | 0.807 | 0.491 | 0.942 | 0.803 |

Abbreviations: AIP, Atherogenic Index of Plasma; BMI, Body Mass Index; B.P, Blood Pressure; CHG Index, Cholesterol, High-density lipoprotein, and Glucose Index; Chol., Cholesterol; HbA1c, Hemoglobin A1c; HDL, High-Density Lipoprotein; HOMA-IR, Homeostatic Model Assessment of Insulin Resistance; LAP, Lipid Accumulation Product; LDL, Low-Density Lipoprotein; MetS, Metabolic Syndrome; TG, Triglycerides; TG/HDL, Triglyceride to High-Density Lipoprotein ratio; TyG Index, Triglyceride-Glucose Index; VAI, Visceral Adiposity Index; WC, Waist Circumference; WHR, Waist-to-Hip Ratio.

association among biochemical markers (adjusted OR: 4.93), WC was the most discriminative among anthropometric measures (adjusted OR: 3.87), and the LAP emerged as the strongest combined index (adjusted OR: 14.91), outperforming all other measurements. Central obesity was the most predominant MetS component in our study, affecting 59% of the overall population.

Our study revealed a MetS prevalence of 19.1% in the cohort, which is lower than previously reported rates. To elaborate, Aqel et al. found a 26% prevalence in their meta-analysis of 14,772 participants, while Al-Thani et al. reported a 28% prevalence in a national health survey on 2,496 Qatari citizens [5,6]. Likewise, Syed et al. reported a higher prevalence of 48.8% in their large cross-sectional study of 127,941 participants [7]. These discrepancies might be partially attributed to our relatively smaller sample size of 692 participants. Moreover, the considerable difference between our findings and those of Syed et al. could be partially explained by methodological differences in the diagnostical criteria. Syde et al. used the NCEP ATP III criteria, while we used the IDF criteria. Nevertheless, the reported prevalence in our study remains slightly low compared to other IDF-based studies suggesting that additional factors may be influencing these differences.

Our study revealed distinct differences in the biochemical profiles of healthy individuals and those with MetS. Specifically, levels of glucose, insulin, HbA1c, total cholesterol, LDL cholesterol, and TG were significantly elevated in participants with MetS, whereas HDL cholesterol levels were significantly reduced. TG showed the strongest association with MetS among the biochemical indices studied (adjusted OR: 4.93, 95% CI: 3.79–6.52), followed by insulin (adjusted

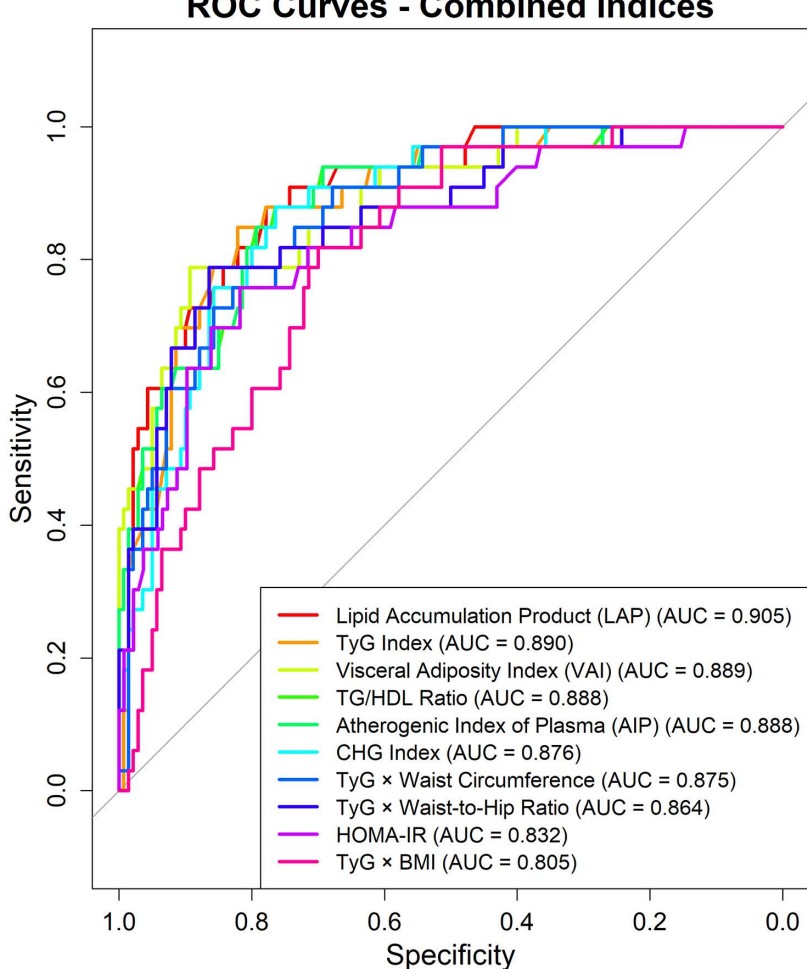

**Fig 4. ROC curves for combined indices on the test dataset.**

OR: 2.19, 95% CI: 1.54–3.28) and HbA1c (adjusted OR: 2.09, 95% CI: 1.77–2.48). These findings align with previous research [28]. In particular, a 2021 cross-sectional study conducted on 381 participants with MetS from Southwest Ethiopia, found a predominant dyslipidemia prevalence in the cohort with specific abnormalities in TG (44.6%) and HDL cholesterol (67.2%) being particularly prevalent, along with significant positive associations between TG and other MetS components, including fasting blood glucose (r = 0.27, p < 0.001), supporting our findings of strong metabolic interconnections [29]. The observed abnormalities in lipid metabolism, evidenced by elevated lipid profiles and reduced HDL cholesterol levels in MetS participants, indicate significant dyslipidemia. Of particular importance, this dyslipidemic state seems to be closely connected with insulin resistance, as evidenced by the concurrent elevations in glucose, insulin, and HbA1c levels. These are two key components in the development and progression of metabolic syndrome. The particularly strong association between TG and MetS (adjusted OR: 4.93) likely mirrors the underlying pathophysiology of metabolic impairment, where elevated TG play a pivotal role as both a marker and mediator of insulin resistance and inflammation through multiple mechanisms, including increased fatty acid flux and lipotoxicity. This bidirectional connection between lipid abnormalities and insulin resistance creates a reinforcing loop that drives the progression of metabolic syndrome, emphasizing the multifaceted nature of metabolic disturbances in this condition.

Central obesity emerged as the most predominant MetS component in our study, affecting 59% of the overall population. This high prevalence highlights the importance of anthropometric measurements in MetS identification. WC demonstrated the strongest discriminative power among anthropometric measures, with its association strengthening after adjustment for demographic factors (adjusted OR: 3.87, 95% CI: 3.10–4.91), followed by WHR (adjusted OR: 3.16, 95% CI: 2.56–3.95) and BMI (adjusted OR: 2.16, 95% CI: 1.82–2.58). The superior discriminative value of WC and WHR, compared to BMI, reflects their ability to specifically capture the distribution of body fat, particularly visceral adipose tissue in the abdominal region, while BMI only provides a general measure of overall adiposity [30–32]. This distinction is crucial because visceral adipose tissue functions as an active endocrine organ, secreting various bioactive substances including inflammatory cytokines (such as TNF-α and IL-6), adipokines, and free fatty acids that directly influence metabolic homeostasis [33].

The use of anthropometric indices to predict the incidence of MetS has been extensively studied [6]. Our findings, which demonstrated WC as the strongest discriminator (AUC 0.816; 95% CI: 0.781–0.85), align with but show notable variations from previous studies. Al-Thani et al. similarly identified WC as the optimal anthropometric discriminator of MetS, though they reported substantially lower discriminative power, with AUC values of 0.612 in men (classified as "poor" performance) and 0.722 in women (classified as "fair" performance) (Al-Thani et al., 2016). Another large-scale study demonstrated a strong association between high WC and MetS (OR 20.07; 95% CI: 19.45–20.71 for ATP III criteria), highlighting that WC is the central essential component for detecting insulin resistance and consequently, the early detection of metabolic syndrome. However, these findings were not demographically adjusted, potentially overstating the relationship [34]. Studies across different age groups have shown varying results. A study on Thai adolescents reported comparable but notably higher discriminative performance for WHR, WC, BMI, and BMI $z$-score (AUC 0.924–0.960), with WHR being more consistent across different age groups [35]. Similarly, a cross-sectional study on the elderly population (60−92 years), reported WHR as the strongest discriminator of MetS with a discriminative ability of (AUC 0.786; 95% CI: 0.76–0.81) [35]. These variations from our findings in a younger population (mean age 42 years) suggest age-related effects on anthropometric indices' discriminative power [36]. The mentioned studies consistently identify WC as a major discriminator for MetS screening, though its discriminative power varies significantly across populations and age groups. The range in the discriminative ability – from the moderate performance in our study (AUC 0.816) to higher values in adolescents (AUC 0.924–0.960) and distinct patterns in the elderly – reflects the complex interplay between aging, body composition, and metabolic risk. These findings highlight the importance of developing age- and population-specific approaches for MetS screening. A particularly noteworthy finding from our study was the significantly higher discriminatory ability of all indices in individuals younger than 45 years. This age-related difference may be attributable to several factors. In older adults (≥45 years), the metabolic landscape is often complicated by a higher prevalence of comorbidities and the use of medications (e.g., statins, antihypertensives), which can alter lipid and glucose levels, thereby reducing the diagnostic clarity of these indices. Furthermore, physiological changes associated with aging, such as alterations in body composition and fat distribution, might weaken the association between these surrogate markers and MetS. This finding suggests that while these indices are exceptionally useful for identifying at-risk younger adults, their interpretation in older populations may require greater clinical nuance and consideration of confounding factors.

The superiority of combined indices over individual indices in predicting MetS could be attributed to their ability to comprehensively assess the various metabolic pathways involved in the development of MetS [37,38]. These indices capture the magnitude of major metabolic dysfunctions, such as insulin resistance, lipid metabolism dysregulation, and inflammation, which are characteristics of metabolic syndrome. Among these, LAP and VAI are recognized as reliable indicators of visceral fat accumulation with a good ability to reflect metabolic health [39]. By incorporating both anthropometric measurements and lipid profiles, these indices provide a more comprehensive assessment of visceral fat. VAI is calculated using WC, BMI, TG, and HDL cholesterol levels, indicating the interplay between visceral fat and lipid abnormalities [40]. In contrast, LAP is a surrogate marker for visceral adiposity that quantifies lipid overaccumulation. It is derived from a

combination of WC and fasting triglyceride levels, reflecting the degree of lipid storage in visceral fat [41]. Other combined measurements such as HOMA-IR and TyG, and the CHG index are also strong discriminators of insulin resistance which is another cornerstone of MetS pathophysiology [42,43]. HOMA-IR and TyG specifically reflect insulin resistance, another cornerstone of MetS pathophysiology, while the CHG index, a novel marker combining total cholesterol, HDL, and glucose, captures broader dyslipidemia and glycemic dysregulation. HOMA-IR, derived from fasting plasma glucose and insulin measurements, captures the dual feedback relationship between these parameters, where glucose stimulates insulin secretion and insulin regulates glucose levels. Moreover, HOMA-IR reflects both hepatic glucose production and pancreatic β-cell function, making it a more comprehensive measure than fasting glucose or insulin alone [44]. The TyG index, calculated from fasting glucose and triglyceride levels, reflects pancreatic function and tissue insulin resistance through glucose-lipid metabolism pathways [45]. TyG has been confirmed to be a strong discriminator of insulin resistance when compared to the gold-standard hyperinsulinemic-euglycemic clamp technique [46].

Combined markers such as LAP, VAI, and TyG have been extensively studied for predicting various MetS components. LAP and VAI have been shown to predict diabetes, pre-diabetes, and cardiovascular mortality [47,48]. The positive association between VAI and cardiometabolic disturbances observed in out cohort is consistent with the systematic review including 32 studies encompassing over 60,000 individuals across diverse age groups and regions, concluding that individuals with elevated VAI consistently exhibited higher blood pressure levels, independent of sex and age, underscoring VAI's robustness as a marker of visceral adiposity and hemodynamic stress [49]. These findings reinforce our interpretation that VAI effectively captures the interplay between adipose tissue dysfunction and cardiometabolic risk. Complementary evidence from an Iranian population also highlighted the significance of VAI among adiposity indices. In their analysis of 9,704 adults from the MASHAD study, VAI remained significantly associated with hypertension in both men and women (OR = 1.03, 95% CI 1.02–1.04), even after adjustment for confounders [50]. Their findings reinforce the importance of VAI as a meaningful anthropometric indicator of blood pressure elevation and metabolic dysregulation across ethnic groups, supporting its inclusion in MetS risk assessment frameworks such as ours.

Moreover, elevated TyG index has been consistently associated with an increased risk of hypertension and cardiovascular disease including coronary artery calcification and arterial stiffness [51,52]. Combined indices demonstrate robust discriminative capabilities, with the combination of functional (biochemical) and structural (anthropometric) measurements reflecting both underlying pathophysiological mechanisms and clinical manifestations. Consequently, combined indices not only provide a comprehensive view of metabolic disturbances but also offer superior clinical utility over individual parameters, enhancing decision-making in MetS assessment.

Our study demonstrated that the combined indices, LAP (AUC = 0.896), VAI (AUC = 0.877), and TyG × WC (AUC = 0.872), had the highest discriminative ability for MetS, with LAP emerging as the strongest discriminator and most balanced performance (sensitivity: 0.856, specificity: 0.789) at a threshold of 37.100. The CHG Index also showed strong performance (AUC = 0.842), positioning it as another valuable, albeit secondary, tool for MetS identification. Although there are no reports of the discriminative ability of these indices in the Qatari population, our findings align with Shin et al., who reported the superiority of LAP among other indices with a slightly higher discriminative ability of (AUC = 0.917) and comparable sensitivity (0.867) and specificity (0.826) in the Korean population [53]. Similarly, Duan et al. used three different diagnostic criteria and reported a range of LAP discriminative abilities ranging from 0.893 to 0.925 in the Chinese population [54]. The optimal LAP threshold identified in our study (37.100) is within the range reported across Asian populations, being comparable to the findings of the study by Shao et al. on the Chinese general population (36.25 for males, 34.95 for females) but slightly higher than the Korean study (33.97). Interestingly, certain populations reported higher discriminative power. For instance, a study by Mosad et al. on a Sudanese cohort reported LAP AUCs of 0.970 and 0.964 for males and females, respectively [55]. This geographic variation in discriminative performance suggests potential ethnic or population-specific factors influencing the relationship between lipid accumulation and MetS development. Gender-specific analyses across studies revealed consistently different optimal thresholds for males and females.

This was particularly evident in Ching et al.'s study of Malaysian vegetarians, where markedly different cutoff values were identified for males (41.435) and females (21.743) [56]. These findings suggest the potential need for gender-specific thresholds in clinical applications, though our study's single threshold of 37.100 demonstrated robust performance across both genders.

While our results confirm LAP's superior performance in general populations, important variations emerged in specific clinical conditions. Notably, Fahmy et al.'s study in chronic kidney disease patients demonstrated LAP's sustained discriminative ability (AUC = 0.916 in males, 0.885 in females) while also highlighting its correlation with insulin resistance [57]. Similarly, Saxena et al. found that VAI outperformed LAP in women with Polycystic ovary syndrome (PCOS), achieving higher sensitivity (88.9%) and specificity (90.7%) [58].

The strong performance of TyG × WC (AUC = 0.872) in our study adds to the growing body of evidence supporting triglyceride and glucose-based indices as valuable markers for MetS identification. However, its consistently lower discriminatory ability compared to LAP across studies suggests it may be more suitable as an alternative marker when LAP calculation is not feasible.

The findings of our study have important clinical implications for the screening and early management of MetS in Qatar. While biochemical and anthropometric measures are widely used due to their convenience, LAP emerged as the strongest predictor, with a sensitivity of 0.856 and specificity of 0.789 at a threshold of 37.100, making it a highly effective and accessible clinical tool. Alternatively, WC alone showed reasonable discriminatory value (AUC 0.816), offering a simple and cost-effective initial screening method. Thus, implementing a two-step screening method – starting with basic anthropometric measurements for initial screening, followed by evaluation of LAP in high-risk individuals – is an effective and practical approach. This strategy balances simplicity and precision, enabling early detection, timely intervention, and improved management of MetS, thereafter reducing its burden.

Our study is the first study to examine the discriminative ability of biochemical, anthropometric, and combined indices for MetS detection in the Qatari population and to define the optimal thresholds for the detection. Secondly, this study examined various indices and included participants with comprehensive and precisely collected metabolic data. Lastly, the association between the different variables and MetS has been comprehensively evaluated statistically by incorporating both adjusted and unadjusted models. This study, however, has some limitations. One limitation is the relatively small sample size. A further limitation is the potential for selection bias. Participants in the QBB are volunteers and may be healthier or more health-conscious than the general population. The cohort also predominantly comprises Qatari citizens, which may limit the generalizability of our findings to the large non-Qatari resident population in the country. Therefore, the prevalence and optimal thresholds of these indices should be validated in more diverse and representative community-based samples from Qatar and the wider Gulf region.

Furthermore, the cross-sectional design prevents the establishment of causal relationships between MetS and the different studied variables. Finally, given that MetS is influenced by multiple factors including ethnicity, genetic predisposition, environmental conditions, and lifestyle variables, the population-specific nature of this study may limit its external validity and applicability to other settings.

In conclusion, this study highlights the superior discriminative capability of combined indices, particularly LAP, for identifying MetS in the Qatari population. While traditional measures like WC offer reasonable initial screening utility, LAP provides a more robust and balanced diagnostic performance. Our findings underscore the importance of incorporating LAP into clinical practice for improved MetS detection and management. The clinical implementation of LAP is highly feasible, particularly in primary care settings. It requires only two common measurements: WC, which is a simple anthropometric measure, and a fasting triglyceride level, which is part of a standard lipid panel. Given its simple calculation and high diagnostic accuracy demonstrated in our study, LAP represents a cost-effective and powerful tool that can be easily integrated into routine health check-ups in Qatar to improve early detection of MetS. Further research in larger, more diverse cohorts is warranted to validate these findings and refine risk stratification strategies.

## Supporting information

**S1 Fig. Correlation matrix plots for biochemical indices.**
(TIF)

**S2 Fig. Correlation matrix plots for anthropometric indices.**
(TIF)

**S3 Fig. Correlation matrix plots for combined indices.**
(TIF)

**S4 Fig. ROC curves for biochemical indices on the test dataset.**
(TIF)

**S5 Fig. ROC curves anthropometric indices on the test dataset.**
(TIF)

**S1 Table. Statistically significant pairwise comparisons of AUC values using DeLong's test.**
(DOCX)

## Acknowledgments

The authors gratefully acknowledge QBB for providing the data used in this research. We extend our appreciation to the data management team for their support and collaboration.

## Author contributions

**Conceptualization:** Muhammad Ammar Zahid, Abdelhamid Kerkadi, Abdelali Agouni.

**Data curation:** Abrar Abdelrahman, Hicham Raïq, Abdelhamid Kerkadi, Abdelali Agouni.

**Formal analysis:** Muhammad Ammar Zahid, Abrar Abdelrahman, Hicham Raïq, Abdelali Agouni.

**Funding acquisition:** Abdelali Agouni.

**Investigation:** Muhammad Ammar Zahid.

**Methodology:** Muhammad Ammar Zahid, Abrar Abdelrahman, Hicham Raïq, Abdelhamid Kerkadi, Abdelali Agouni.

**Project administration:** Abdelhamid Kerkadi, Abdelali Agouni.

**Resources:** Abdelali Agouni.

**Software:** Muhammad Ammar Zahid, Hicham Raïq.

**Supervision:** Hicham Raïq, Abdelali Agouni.

**Validation:** Muhammad Ammar Zahid, Abrar Abdelrahman, Abdelhamid Kerkadi, Abdelali Agouni.

**Visualization:** Muhammad Ammar Zahid, Abrar Abdelrahman.

**Writing – original draft:** Muhammad Ammar Zahid, Abrar Abdelrahman, Abdelali Agouni.

**Writing – review & editing:** Muhammad Ammar Zahid, Abrar Abdelrahman, Hicham Raïq, Abdelhamid Kerkadi, Abdelali Agouni.

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
