## [Decision Letter · Decision Letter 0]

29 Sep 2025

Dear Dr. Agouni,

Thank you for submitting your manuscript to PLOS ONE. After careful consideration, we feel that it has merit but does not fully meet PLOS ONE’s publication criteria as it currently stands. Therefore, we invite you to submit a revised version of the manuscript that addresses the points raised during the review process.

Dear Author

As you will note, three experts have reviewed the manuscript, and all have raised significant concerns regarding aspects such as the novelty, grammar, and other issues. Nevertheless, I believe the authors should be given the opportunity to revise the manuscript in response to the reviewers' comments.

Best regards,

Amin Mansoori

We look forward to receiving your revised manuscript.

Kind regards,

Amin Mansoori

Academic Editor

PLOS ONE

Journal Requirements:

2. In the online submission form, you indicated that all relevant data are included within the manuscript. Additional data or information can be provided by the corresponding author upon reasonable request.

The study was funded by the Qatar National Research Fund (Qatar Research Development and Innovation Council) [grant No. NPRP14S-0406-210150] and Qatar University [grants No. QUST-1-CPH-2025-247 and QUST-2-CPH-2021-221]. M.A.Z. is supported by a Ph.D. graduate assistantship from the Office of Graduate Studies (Qatar University). The statements made herein are solely the responsibility of the authors.

The study was funded by the Qatar National Research Fund (Qatar Research Development and Innovation Council) [grant No. NPRP14S-0406-210150] and Qatar University [grants No. QUST-1-CPH-2025-247 and QUST-2-CPH-2021-221]. M.A.Z. is supported by a Ph.D. graduate assistantship from the Office of Graduate Studies (Qatar University). The statements made herein are solely the responsibility of the authors.

Additional Editor Comments 

Dear Author

As you will note, three experts have reviewed the manuscript, and all have raised significant concerns regarding aspects such as the novelty, grammar, and other issues. Nevertheless, I believe the authors should be given the opportunity to revise the manuscript in response to the reviewers' comments.

Best regards,

Amin Mansoori

Reviewers' comments:

Reviewer's Responses to Questions

**Comments to the Author**

1. Is the manuscript technically sound, and do the data support the conclusions?

Reviewer #1: Yes

Reviewer #2: Yes

Reviewer #3: Partly

2. Has the statistical analysis been performed appropriately and rigorously?

Reviewer #1: Yes

Reviewer #2: Yes

Reviewer #3: Yes

3. Have the authors made all data underlying the findings in their manuscript fully available?

Reviewer #1: Yes

Reviewer #2: No

Reviewer #3: No

4. Is the manuscript presented in an intelligible fashion and written in standard English?

Reviewer #1: Yes

Reviewer #2: Yes

Reviewer #3: Yes

Reviewer #1: Referee's opinion on the paper,

“Assessing the diagnostic accuracy of biochemical, anthropometric, and combined indices for metabolic syndrome prediction in a cohort from Qatar Biobank” submitted in “PLOS ONE”.

The manuscript investigated the comparison of the diagnostic accuracy of various indices for MetS prediction in QBB. The study was on 692 individuals and cross sectional. Authors used LR method to analyze the data. The methodology of the manuscript is enough however, I have some comments which are given as follows:

1- Introduction: Some sentences need references. For example, “Various criteria exist for diagnosing MetS, with the 2009 Harmonized Criteria being the most widely accepted.”, “A 2024 systematic review and meta-analysis reported an overall pooled prevalence of 26% in the Qatari population, with notable differences across age groups and diagnostic criteria.”, and … .

2- Introduction is not well-structured. It is so long and need to be shorten.

3- Abbreviation should be defined in first seen. Check whole paper in this regard. For example, QBB many times defined.

4- Do the data have any information about the medications of participants? It is an important factor related to MetS and Diabetes.

5- Discussion section needs more comparable explanation with literatures. For example, abdominal volume index (VAI) (10.3389/fcvm.2024.1341229, 10.1016/j.orcp.2021.10.001, 10.1038/s41371-023-00877-z, 10.1016/j.numecd.2021.04.024) is another index that can be compared or even adding in the analysis of the manuscript.

6- Recently, Cholesterol, High density lipoprotein, and Glucose (CHG) index (doi: 10.1111/jdi.14343) for T2DM was introduced that more efficient than TyG. I suggest adding this in your analysis.

Reviewer #2: The manuscript “Assessing the diagnostic accuracy of biochemical, anthropometric, and combined indices for metabolic syndrome prediction in a cohort from Qatar Biobank” (PONE-D-25-08890) presents a well-conducted cross-sectional analysis that evaluates diagnostic indices for MetS in a Qatari cohort.

The study addresses an important gap, given the high prevalence of metabolic risk factors in Qatar, and provides useful evidence on the comparative performance of biochemical, anthropometric, and combined indices.

The study has several strengths:

- The rationale and objectives are clearly presented, the sample size is adequate (n=692), and standardized IDF criteria were used.

- The methodology is robust, employing logistic regression, ROC analysis, DeLong’s test, stratified analyses, and internal validation through a train-test split.

- The reporting of results is clear, with tables and figures that effectively display comparisons across indices.

- A key strength is the identification of LAP as the strongest predictor (AUC=0.896), with practical diagnostic thresholds that could be useful in clinical practice.

At the same time, there are concerns that need to be addressed before the manuscript can be considered for publication.

- The cross-sectional design restricts inference to associations at a single point in time. The manuscript should make clearer that the term “predictive performance” refers to discriminatory ability in a cross-sectional context, not prediction of future risk.

- The representativeness of the Qatar Biobank sample is also an issue, since participants may not reflect the broader Qatari or regional population. This potential selection bias requires explicit discussion.

- Another key issue is the clinical applicability of LAP and combined indices. Although they outperform traditional measures, the authors should explain whether these indices are easily calculated and practical for routine clinical use in Qatar.

- The age-stratified results are noteworthy, showing markedly higher predictive accuracy in younger participants (<45 years). This important finding is not fully explored. The authors should discuss possible explanations, such as the higher prevalence of co-morbidities or confounding factors in older participants, and whether this limits generalizability of the indices to older adults.

- The data availability statement also requires revision. PLOS ONE requires that data be openly accessible at the time of publication. The current phrasing (“available upon request”) is not compliant, and the authors should either deposit the data in a repository or provide a stronger justification for restrictions.

Minor issues include:

- Check for typographical errors (e.g., “Figurte S1”), also, some redundancy in the Introduction where the burden of MetS is repeated, and the need for clearer reporting of odds ratios.

- Since continuous variables were standardized, this should be stated explicitly to aid interpretation.

- Tables could also be simplified by merging overlapping content and providing clearer labeling for the combined indices.

- The discussion could benefit from comparisons with similar regional studies beyond Qatar to highlight external validity.

Overall, this is a strong and well-structured manuscript with clear clinical relevance. However, revisions are needed to strengthen the discussion of generalizability, clarify the implications of the age-stratified results, address clinical feasibility, and ensure compliance with data availability requirements.

Therefore, I recommend major revision.

Reviewer #3: This is a comprehensive manuscript that addresses a critical public health problem. However, the validity and interpretation of the results are difficult to assess due to a lack of clarity in the presentation. Many of the figures appear unclear, blurry, and ambiguous, which hinders the reader’s ability to interpret the data effectively. Additionally, I noticed some typographical errors, such as on line 155, which should be corrected. I recommend that the authors revise the manuscript for clarity, improve figure quality, and carefully proofread the text to enhance readability and ensure accurate communication of the findings.

**Do you want your identity to be public for this peer review?** For information about this choice, including consent withdrawal, please see our Privacy Policy

Reviewer #1: No

Reviewer #2: **Yes: ** Dalal Usamah Zaid Alkazemi

Reviewer #3: No

---

## [Author Response · Author response to Decision Letter 1]

7 Oct 2025

Point-by-Point Responses to the Editor and Reviewers’ Comments

EDITOR / JOURNAL REQUIREMENTS

1. Ensure manuscript meets PLOS ONE style requirements.

Response: Manuscript re-formatted per PLOS ONE style: section headings, figure/table numbering, and file naming have been standardized.

2 & 3. Data-availability compliance and justification.

Response: The dataset is owned by Qatar Biobank (QBB) and contains identifiable participant data. Public release would breach MTA, NDA, and IRB terms and conditions.

Inserted statement:

“Data are available from Qatar Biobank under a controlled-access process and cannot be publicly shared due to IRB and data-transfer restrictions (IRB: Ex-2020-RES-ACC-0215-0125; Ex-2021-QF-QBB-RES-ACC-00049-0173). Requests should be directed to qphi@qf.org.qa.”

4. Funding statement placement.

Response: The following text was retained only in the Funding Statement section and deleted from Acknowledgments:

The study was funded by the Qatar National Research Fund (Qatar Research Development and Innovation Council) [grant No. NPRP14S-0406-210150] and Qatar University [grants No. QUT2RP-CPH-24/25-477, QUST-1-CPH-2025-247, and QUST-2-CPH-2021-221]. M.A.Z. is supported by a Ph.D. graduate assistantship from the Office of Graduate Studies (Qatar University). The funders had no role in study design, data collection and analysis, decision to publish, or preparation of the manuscript.

5. Abstract consistency.

Response: Abstract on submission form and in manuscript now identical.

6. Evaluation of suggested literature.

Response: All reference papers were reviewed, and relevant ones cited in the Discussion.

REVIEWER #1

1. Missing references in the Introduction.

Response: We have now included supporting citations for all unreferenced claims.

2. Introduction too long / restructure.

Response: The section was condensed and reorganized for logical flow; redundant epidemiological detail was removed.

3. Abbreviations not defined at first appearance.

Response: Full manuscript checked—each abbreviation defined at first use.

4. Medication data missing.

Response: A lot of information on medication is missing in our dataset, so we could not, unfortunately, include this information in the analysis.

5. Discussion lacks a comparable explanation (VAI).

Response: We have included a comparative discussion. The inserted paragraph includes:

“The positive association between VAI and cardiometabolic disturbances observed in out cohort is consistent with the systematic review including 32 studies encompassing over 60,000 individuals across diverse age groups and regions, concluding that individuals with elevated VAI consistently exhibited higher blood pressure levels, independent of sex and age, underscoring VAI’s robustness as a marker of visceral adiposity and hemodynamic stress [49]. These findings reinforce our interpretation that VAI effectively captures the interplay between adipose tissue dysfunction and cardiometabolic risk. Complementary evidence from an Iranian population also highlighted the significance of VAI among adiposity indices. In their analysis of 9,704 adults from the MASHAD study, VAI remained significantly associated with hypertension in both men and women (OR = 1.03, 95% CI 1.02–1.04), even after adjustment for confounders [50]. Their findings reinforce the importance of VAI as a meaningful anthropometric indicator of blood pressure elevation and metabolic dysregulation across ethnic groups, supporting its inclusion in MetS risk assessment frameworks such as ours.”

6. Add CHG index.

Response: The CHG index is now incorporated into methods, analysis, and results tables. We added the following line in the Introduction:

“More recently Cholesterol, High-density lipoprotein, and Glucose (CHG) index have been used for the diagnosis of diabetes mellitus and is associated with MetS [19,20].”

REVIEWER #2

1. Clarify that “predictive performance” refers to cross-sectional discrimination.

Response: We replaced predictive-related terminology with “discriminatory ability/performance” throughout.

2. Representativeness of Qatar Biobank sample.

Response: We added a limitation paragraph discussing QBB volunteer bias and limited generalizability.

“A further limitation is the potential for selection bias. Participants in the Qatar Biobank are volunteers and may be healthier or more health-conscious than the general population. The cohort also predominantly comprises Qatari citizens, which may limit the generalizability of our findings to the large non-Qatari resident population in the country. Therefore, the prevalence and optimal thresholds of these indices should be validated in more diverse and representative community-based samples from Qatar and the wider Gulf region.”

3. Clinical applicability of LAP

Response: We added a discussion paragraph as follows:

“The clinical implementation of LAP is highly feasible, particularly in primary care settings. LAP requires only two common measurements: waist circumference, which is a simple anthropometric measure, and a fasting triglyceride level, which is part of a standard lipid panel. Given its simple calculation and high diagnostic accuracy demonstrated in our study, LAP and VAI represent a cost-effective and powerful tool that can be easily integrated into routine health check-ups in Qatar to improve early detection of MetS.”

4. Age-stratified results.

Response: We added an explanation for stronger performance in younger participants as follows:

“A particularly noteworthy finding from our study was the significantly higher discriminatory ability of all indices in individuals younger than 45 years. This age-related difference may be attributable to several factors. In older adults (≥45 years), the metabolic landscape is often complicated by a higher prevalence of comorbidities and the use of medications (e.g., statins, antihypertensives), which can alter lipid and glucose levels, thereby reducing the diagnostic clarity of these indices. Furthermore, physiological changes associated with aging, such as alterations in body composition and fat distribution, might weaken the association between these surrogate markers and MetS. This finding suggests that while these indices are exceptionally useful for identifying at-risk younger adults, their interpretation in older populations may require greater clinical nuance and consideration of confounding factors.”

5. Data-availability revision.

Response: We updated the statement as follows:

“Data are available from Qatar Biobank under a controlled-access process and cannot be publicly shared due to IRB and data-transfer restrictions (IRB: Ex-2020-RES-ACC-0215-0125; Ex-2021-QF-QBB-RES-ACC-00049-0173). Requests should be directed to qphi@qf.org.qa.”

6. Typographical and redundancy corrections.

Response: All typos have been corrected, and redundant statements have been removed.

7. Continuous variables standardized.

Response: We added the following statement to the Methods: “For the logistic regression analysis, all continuous variables were standardized (converted to z-scores) to allow for the comparison of odds ratios across different measures.”

8. Simplify tables / clarify labels.

Response: We included abbreviation footnotes and uniform labeling for all tables.

REVIEWER #3

1. Figures unclear/blurry.

Response: We replaced those figures with high-resolution versions in corrected aspect ratios. All relevant information is also presented in the associated tables.

2. Typographical errors.

Response: We proofread the entire text; all typographical issues were now corrected.

---

## [Decision Letter · Decision Letter 1]

26 Oct 2025

Dear Dr. Agouni,

Thank you for submitting your manuscript to PLOS ONE. After careful consideration, we feel that it has merit but does not fully meet PLOS ONE’s publication criteria as it currently stands. Therefore, we invite you to submit a revised version of the manuscript that addresses the points raised during the review process.

We look forward to receiving your revised manuscript.

Kind regards,

Amin Mansoori

Academic Editor

PLOS ONE

Journal Requirements:

Reviewers' comments:

Reviewer's Responses to Questions

**Comments to the Author**

Reviewer #2: All comments have been addressed

Reviewer #3: (No Response)

2. Is the manuscript technically sound, and do the data support the conclusions?

Reviewer #2: Yes

Reviewer #3: Yes

3. Has the statistical analysis been performed appropriately and rigorously?

Reviewer #2: Yes

Reviewer #3: Yes

4. Have the authors made all data underlying the findings in their manuscript fully available?

Reviewer #2: Yes

Reviewer #3: Yes

5. Is the manuscript presented in an intelligible fashion and written in standard English?

Reviewer #2: Yes

Reviewer #3: Yes

Reviewer #2: The authors added the CHG index, improved the Introduction, clarified terminology, added missing references, and corrected tables and figures. Discussion sections now include comparisons for VAI and LAP, along with explanations of age effects and clinical relevance.

The manuscript is clear, complete, and ready for acceptance.

Reviewer #3: The figures are hard to read and correlate to the note written. Can the figures be made more readable? Also, why not have the figures where the legend is located just like the tables? That would make the reading flow better even for reviewers.

**Do you want your identity to be public for this peer review?** For information about this choice, including consent withdrawal, please see our Privacy Policy

Reviewer #2: **Yes: ** Dalal Usamah Zaid Alkazemi

Reviewer #3: No

---

## [Author Response · Author response to Decision Letter 2]

29 Oct 2025

REVIEWER #3

1. The figures are hard to read and correlate to the note written. Can the figures be made more readable? Also, why not have the figures where the legend is located just like the tables? That would make the reading flow better even for reviewers.

Response: We thank the reviewer for their feedback. We have addressed the two points as follows:

• Figure Readability: This issue has been fully resolved. We have replaced all figures with high-resolution versions, ensuring each one complies strictly with the journal’s technical specifications, including a minimum resolution of 300 DPI and adherence to the maximum dimensions of 7.5 x 8.25 inches. The revised figures are now crisp, clear, and fully legible.

• Figure Placement: Regarding the suggestion to embed figures within the text alongside their captions, this request is in direct conflict with PLOS ONE's mandatory author guidelines. The journal's formatting policy explicitly requires authors to submit figure files separately and not to embed them in the manuscript file. The guidelines state that captions must be listed in the manuscript text after the first paragraph of their citation.

Accordingly, we have strictly adhered to the journal’s submission protocol. To do otherwise would be a direct violation of the submission requirements. We are confident that with the vastly improved resolution, correlating the figures with their corresponding captions in the text is now straightforward. We trust this clarifies why we could not accommodate this specific suggestion and confirms our compliance with the journal's formatting standards.

---

## [Decision Letter · Decision Letter 2]

7 Dec 2025

Assessing the diagnostic accuracy of biochemical, anthropometric, and combined indices for metabolic syndrome prediction in a cohort from Qatar Biobank

PONE-D-25-08890R2

Dear Dr. Abdelali Agouni,

We’re pleased to inform you that your manuscript has been judged scientifically suitable for publication and will be formally accepted for publication once it meets all outstanding technical requirements.

Kind regards,

Marwan Salih Al-Nimer, MD, PhD

Academic Editor

PLOS One

Additional Editor Comments (optional):

Reviewers' comments:

Reviewer's Responses to Questions

**Comments to the Author**

Reviewer #2: All comments have been addressed

Reviewer #4: (No Response)

2. Is the manuscript technically sound, and do the data support the conclusions?

Reviewer #2: Yes

Reviewer #4: No

3. Has the statistical analysis been performed appropriately and rigorously?

Reviewer #2: Yes

Reviewer #4: No

4. Have the authors made all data underlying the findings in their manuscript fully available?

Reviewer #2: Yes

Reviewer #4: No

5. Is the manuscript presented in an intelligible fashion and written in standard English?

Reviewer #2: Yes

Reviewer #4: No

Reviewer #2: - Explicitly state whether all models were adjusted for potential confounders such as age and smoking. If not adjusted, note this as a limitation.

Reviewer #4: The authors investigate the association and diagnostic performance of biochemical, anthropometric, and composite indices for predicting metabolic syndrome. Their findings show that variables such as fasting glucose, HbA1c, triglycerides, HDL-C, waist circumference, BMI, and several derived indices are associated with metabolic syndrome and can discriminate individuals with and without the condition.

However, a fundamental conceptual issue arises: several of these parameters—specifically waist circumference, triglycerides, HDL-C, and glucose—are themselves diagnostic components of metabolic syndrome according to widely adopted criteria (e.g., NCEP ATP III, IDF). Evaluating the association or predictive accuracy of variables that directly define the condition introduces circular reasoning. Demonstrating that diagnostic criteria predict the diagnosis they determine does not add scientific value and risks overstating the novelty or clinical utility of the findings.

The authors should explicitly justify why these analyses were performed, clarify the intended incremental value beyond the known diagnostic components, and consider focusing on indices or biomarkers that are not part of the defining criteria of metabolic syndrome. Without this clarification, the study’s conclusions may be interpreted as tautological rather than informative.

**Do you want your identity to be public for this peer review?** For information about this choice, including consent withdrawal, please see our Privacy Policy

Reviewer #2: **Yes: ** Dr. Dalal Usamah Zaid Alkazemi

Reviewer #4: No

---

## [Editor Report · Acceptance letter]

PONE-D-25-08890R2

PLOS One

Dear Dr. Agouni,

I'm pleased to inform you that your manuscript has been deemed suitable for publication in PLOS One. Congratulations! Your manuscript is now being handed over to our production team.

Kind regards,

on behalf of

Professor Marwan Salih Al-Nimer

Academic Editor

PLOS One